# Monitoring the TROPOMI-SWIR module instrument stability using desert sites

Tim A. van Kempen[1], Filippo Oggionni[2], and Richard M. van Hees[1]

[1]SRON Netherlands Institute for Space Research, Sorbonnelaan 2, 3584 CA, Utrecht, the Netherlands
[2]Delft University of Technology, Faculty of Aerospace Engineering, Kluyverweg 1, 2629 HS, Delft, the Netherlands

**Correspondence:** Tim van Kempen (t.a.van.kempen@sron.nl)

**Abstract.** Since its launch in 2017, the TROPOMI instrument on S-5P has provided very high quality data using daily global coverage for a number of key atmospheric trace gasses. Over its first 1,000 days in operations, the SWIR module has been very stable and the continuously monitored calibration has remained of high quality. This calibration relies on a combination of extensive pre-launch and post-launch measurements, complemented by regular monitoring of internal light sources and background measurements. In this paper we present a method and results for independent validation of the SWIR module calibration and instrument stability by examining the signal stability of a sample of 23 pseudo-invariant calibration desert sites. The data covers over two years of operational data. With a Lambertian surface assumption, the results show that the SWIR module has little to no instrument degradation down to an accuracy of about 0.3% per year, validating results obtained from the internal calibration suite. The method presented here will be used as ongoing validation of the SWIR calibration.

## 1 Introduction

The Tropospheric Monitoring Instrument (TROPOMI[1]) is the sole instrument on the Sentinal-5 Precursor mission (better known as S-5P Veefkind et al., 2012). S5-P is part of the ESA Copernicus program and is the first mission covering the atmospheric composition. Its aim is to quantify specific trace-gas column densities. With the very wide swath opening of more than 60 degrees, TROPOMI can provide daily global coverage. The four channels (UV, UVIS, NIR and SWIR) give unprecedent view of atmospheric composition and evolution on timescales of days to years. Launched in late 2017, TROPOMI started regular operations in April 2018 after in-flight commissioning and science verification. The instrument has been shown to be very stable with only limited degradation (van Kempen et al., 2019; Ludewig et al., 2020). The SWIR module covers a wavelength range between 2300 and 2380 nm and has been designed for the carbon-monoxide (CO) and methane ($CH_4$) products, (e.g. Hu et al., 2018; Borsdorff et al., 2018; Pandey et al., 2019; Zhang et al., 2020).

Since the start of nominal operations, TROPOMI has been in excellent health and the calibration has remained stable, in particular the SWIR module (van Kempen et al., 2019; Ludewig et al., 2020). For its calibration, TROPOMI relies on the

---

[1]TROPOMI is a collaboration between Airbus Defence and Space Netherlands, KNMI, SRON and TNO, on behalf of NSO and ESA. Airbus Defence and Space Netherlands is the main contractor for the design, building and testing of the instrument. KNMI and SRON are the principal investigator institutes for the instrument. TROPOMI is funded by the following ministries of the Dutch government: the Ministry of Economic Affairs, the Ministry of Education, Culture and Science, and the Ministry of Infrastructure and the Environment.

results from the on-ground campaign (Kleipool et al., 2018; Tol et al., 2018; van Hees et al., 2018), regular solar irradiance measurements and background measurements, as well as measurements using on-board lights that are part of the integrated calibration unit (ICU) (see van Kempen et al., 2019, for an overview of on-board lights relevant to the SWIR module). The on-board lights consist of a White Light Source (WLS), Dedicated LED (DLED), five diode lasers and a common Led (CLED).The on-board lights have various goals to maintain specific parts of the calibration; e.g. the diode lasers are aimed at characterizing any ISRF (Instrument Spectral Response Function) and/or straylight changes (Tol et al., 2018; van Hees et al., 2018). In-flight, the main goal is the relative calibration, although some parts of the absolute calibration can be derived. The results reveal instrument instabilities for the SWIR module to be less than 0.5% over the first year. This stability has been improved by regular monitoring (see e.g. mps.tropomi.edu), with the SWIR module stable down to 0.3% or less over the first two or three years of regular operations.

TROPOMI products are provided either as L1b products, which consists of calibrated radiance and irradiance data, or L2 products, which consist of the retrieved column densities of selected gas tracers. Data quality from TROPOMI is validated using a range of space-based, airborne and ground-based networks and/or dedicated measurements[2]. These validations are done using the full range of the L2 products. However, these efforts, which are part of the Mission Performance Center of S-5P do not include independent validation of the L1b data quality due to the complexity of the task. Various scenarios have been discussed, but in essence, L1b validation is much more complicated and cannot easily be carried out using automated ground-stations.

Nevertheless, validation of the L1b product serves two important goals. First, it provides validation of the quality of the absolute and relative radiometric calibration and thus the quality of the calibration key data in use. Second, it provides an independent quantification of the instrument stability. The first goal requires independent ground-based measurements. These typically originate from co-incidental space-based measurements (so-called cross-calibration, see e.g. Chander et al., 2013; Kataoka et al., 2017) or from complementary ground measurements through Vicarious Calibration (e.g. Kuze et al., 2014; Bruegge et al., 2019b, a). Both have strict limitations due to instrument properties, temporal difference or spatial coverage. However, validation of instrument stability has less stringent demands and can be carried out by monitoring Pseudo-Invariant Calibration Sites, from hereon referred to as PICS. PICS are locations on Earth that are typically free of precipitation, vegetation and/or clouds. They are relatively free of annual changes (e.g. due to vegetation) and posses a homogeneous surface. Desert locations are typically well suited as PICS. PICS have been in use as calibration sites since the mid-90s (Cosnefroy et al., 1996; Lacherade et al., 2013; Bacour et al., 2019), but are almost exclusively used by CCD camera's in the visible part of the spectrum. Other well-known sites include RailRoad Valley Playa (Nevada, US), Gobabeb (Namibia) or Boatou (China), all part of the RADCAL network (Bouvet et al., 2019).

Note that even in optimal circumstances, annual variations may exist, due to e.g. moisture content or small vegetation, both in the top soil. Non-lambertian reflectance components of the surface is expressed through the Bidirection Reflectance Distribution Function (BRDF) which determines into which directions sunlight is reflected.

---

[2]Please see VDAF facility at http://mpc-vdaf.tropomi.eu for up to date validation activities.

The BRDF of desert sand, in particular its accuracy and effect at off-nadir viewing angles, has been described and charac-
terized by Bruegge et al. (2019a). They find that desert sand has a characteristic ′hot-spot′ in the backwards direction of the
incoming solar radiation. They also show that these BRDFs are not well characterized at large off-nadir viewing angles, with
differences of 5% in a normalized solution w.r.t. the zenith shown as typical. The MODIS BRDF product (Schaaf et al., 2011)
provides world-wide coverage with a 16-day interval but also suffers from inaccuracies as described in Bruegge et al. (2019a)
due to saturation of the detector. These are relevant for TROPOMI, due to its very wide swath.

  In this paper we present an analysis of the site stability of a sample of 23 PICS in the Saharan, Arabian and Namibian
deserts[3]. The sample was adopted from the analysis of Bacour et al. (2019), which itself revisited the original Saharan PICS
sample from Cosnefroy et al. (1996) and Lacherade et al. (2013). Most analyses of PICS sites are done at visible wavelengths
and/or at high resolutions of imagers. In this analysis, we will use continuum channels of the TROPOMI SWIR channel
assuming a Lambertian surface. In addition, no atmospheric correction will be applied. Section 2 provides the final sample,
while the data usage of TROPOMI is presented in Section 3. This includes data filters and restrictions as well as the assumptions
in our analysis. Section 4 gives the results while the impact is discussed in Section 5.

## 2 Saharan sites

Saharan PICS have been used for monitoring for a large number of sensors (e.g. Cosnefroy et al., 1996; Lacherade et al., 2013;
Bacour et al., 2019). These studies have shown the Sahara to be reasonably stable across large spatial scales and relatively
invariant in time. The most recent in-depth review of the spatial and temporal variability of Saharan PICS can be found in
Bacour et al. (2019) where the original lists of Cosnefroy et al. (1996); Lacherade et al. (2013) were inspected. An update
of coordinates was given as compared to the original locations of Cosnefroy et al. (1996), as well as a quantified ordering on
spatial homogeneity across 20 and 100 kilometers as well as temporal stability. In this study we have adopted the final sample
of Bacour et al. (2019) in their Tables 2 and 3. We will be using their coordinates to track the temporal variability of the SWIR
channel continuum signal. As a typical TROPOMI pixel is at least 7x5 km$^2$ but can be as large as 20x7 km$^2$ (especially during
the early parts of operations), the sites are inspected on their temporal stability and spatial homogeneity at 20 km (TVar (20km)
and SHom(20km), as reported in Table 2 of Bacour et al. (2019).

## 3 TROPOMI Data

S5-P has daily overpasses over all sites shown in Table 1. For most sites, multiple overpasses only exist every ∼5 days due to
the proximity to the equator. An overpass is valid if a pixel is located less than 0.2 degrees from the reference location given in
Table 1. At the equator this equals 22 km. At typical Saharan latitudes, the separation is less then 20 km. Note that if multiple
pixels qualify, only the pixel with the smallest separation to the reference position is used. TROPOMI data was processed using
the operational version of the L1b version 01.00.00. Several filters are applied to improve the overall data quality. These are

---

[3]For simplicity, we will refer to this sample as the Saharan PICS sample, even though not all sites are located in the Saharan region

**Table 1.** Alphabetical list of Saharan PICS from Bacour et al. (2019) with latitude longitude following updated coördinates. Time variability and spatial homogeneity are given for reference.

| Site | Latitude | Longitude | Tvar | Svar |
| --- | --- | --- | --- | --- |
| | degree | degree | % | % |
| Algeria1 | 23.83 | -0.76 | 2.1 | 0.3 |
| Algeria2 | 25.66 | -0.62 | 2.0 | 0.7 |
| Algeria3 | 30.63 | 7.83 | 2.0 | 0.7 |
| Algeria4 | 29.99 | 5.10 | 1.8 | 0.5 |
| Algeria5 | 31.16 | 2.24 | 2.1 | 0.8 |
| AlgeriaPICSAND1 | 31.70 | 8.35 | 2.1 | 0.6 |
| Arabia1 | 19.80 | 47.07 | 1.4 | 0.4 |
| Arabia2 | 20.19 | 51.63 | 1.4 | 0.2 |
| Arabia3 | 28.80 | 43.05 | 2.1 | 0.7 |
| ArabiaPICSAND1 | 29.26 | 40.91 | 1.9 | 0.5 |
| Egypt1 | 26.61 | 26.22 | 1.9 | 0.5 |
| Libya1 | 24.65 | 13.25 | 2.1 | 0.6 |
| Libya2 | 25.08 | 20.77 | 2.0 | 0.4 |
| Libya3 | 23.22 | 23.23 | 1.4 | 0.4 |
| Libya4 | 28.67 | 23.42 | 2.1 | 0.6 |
| Mali1 | 19.14 | -5.77 | 2.2 | 0.3 |
| Mauritania1 | 19.51 | -8.57 | 1.9 | 0.6 |
| Mauritania2 | 19.78 | -8.89 | 1.9 | 0.6 |
| NamibiaPICSAND1 | -25.00 | 15.25 | 1.2 | 0.9 |
| Niger1 | 20.26 | 9.64 | 2.1 | 0.3 |
| Niger2 | 21.33 | 10.60 | 2.2 | 0.6 |
| Niger3 | 21.51 | 7.86 | 2.3 | 1.2 |
| Sudan1 | 22.11 | 28.11 | 1.5 | 0.5 |
| SudanPICSAND1 | 17.26 | 25.53 | 1.7 | 0.7 |

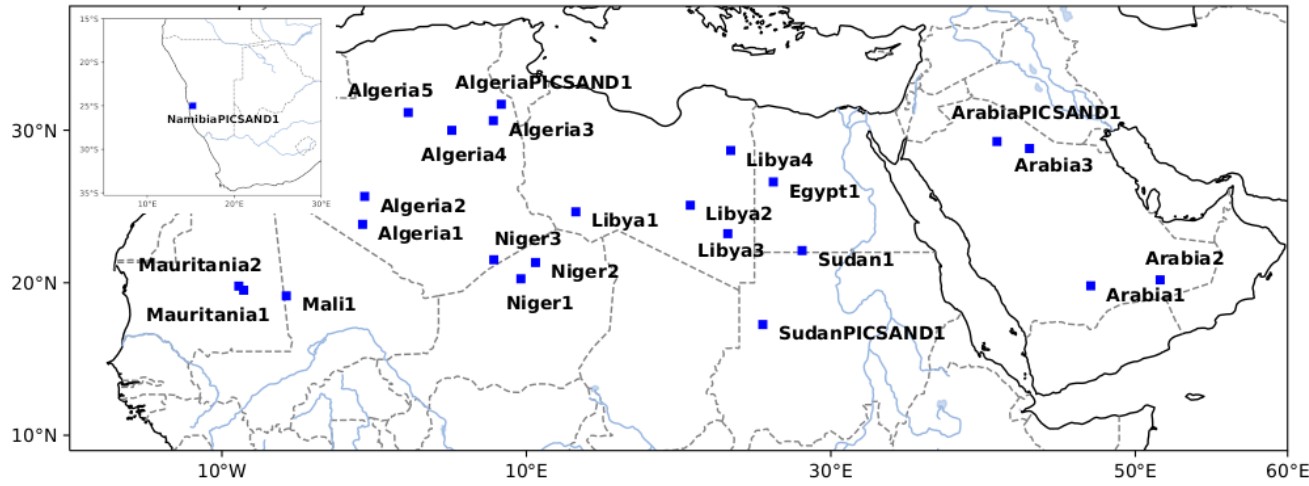

**Figure 1.** Location of Saharan and Arabian PICS. The PICS in the Namibian desert is shown as an inset.

listed in Table 2. Apart from the separation restriction, the viewing zenith angle must be smaller than 50 degrees, the cloud fraction of the pixel must be below 0.02, a successful and valid irradiance measurement should have been taken within a day of the overpass and the solar zenith angle must be below 60 degrees[4]. With the constraints described above, the largest variable parameter is the size of the TROPOMI overpass pixel. The size of a TROPOMI pixel is dependent on its location in the swath. At nadir, a typical TROPOMI pixel at SWIR wavelength is $7 \times 7$ km$^2$, while at the edge pixels can dramatically increase in size in the across-track direction, growing to sizes of up to $7 \times 26$ km$^2$. Note that after August 2019, this was reduced to $5.5 \times 7$ km$^2$ due to shorter integration times. As seen from Bacour et al. (2019), spatial homogeneity at larger scales (i.e. 100 km) can be significantly worse than its counterpart at 20 km scales. As such, it is imperative to avoid 'contamination' from large scales and focus only on the central regions. TROPOMI-SWIR soundings are thus restricted to viewing zenith angles of 50 degrees or less. Effects of larger viewing angles will be investigated in a future paper. Effectively this constrains soundings to positions away from the edges of the swath. This also automatically limits the pixel to a maximum pixel size in the across-track direction, while in the along-track dimension the resolution is set by the sampling time. The pixel size is thus dependent on the location through the viewing angle and on the date due to the change in integration time. In general, pixels are always smaller than 7 km in the along-track and 8.5 km in the across-track.

A choice was made to include relatively large viewing angles (up to 50 degrees) but exclude more extreme angles (TROPOMI can observe at almost 65 degrees). The alternatives, either limits to small viewing zenith angles or including all viewing angles, either severely constrained the available data or introduced additional scatter. Other alternatives, such as binning the data and removing angle-dependencies in the scatter did not improve the data. To compare data over the full time-span, data is corrected

---

[4]Note that this is a general restriction. Due to the early afternoon overpass time of S-5p, all measurements pass this restriction. It is included for completeness in the methodology.

**Table 2.** Constraints used for monitoring of the TROPOMI-SWIR radiance.

| Constraints | |
| --- | --- |
| Overpass separation | <0.2 degree |
| Instrument Zenith Angle | <50 degree |
| Cloud Fraction | <0.02 |
| Irradiance separation | <1 day |
| Solar Zenith Angle | <60 degree |
| Continuum wavelengths | [2312.7,2312.9] nm |

by dividing by the cosine of the solar zenith angle. An additional constraint to limit the maximum solar zenith angle was implemented to filter a few problematic cases during winter times.

Cloud fraction is defined as a number between 0 and 1 representing the area of a pixel covered by clouds. A number of 0 is completely cloud-free, while 1 represents a pixel completely covered by clouds. The cloud fraction information was extracted from the support data included in the operational data product of the $CH_4$ ′offline′processor, version 1.2.0. Given the straylight correction as defined in Tol et al. (2018) and applied in the processor, no filtering was used for cloud presence beyond the sounding pixel.

Last but not least, soundings are required to have a valid irradiance measurement using TROPOMI-SWIR within a day. This filter is aimed at possible problematic soundings taken after an orbit maneuver or anomaly. This affects less than 0.1% of all TROPOMI measurements.

    If an overpass is considered valid, the radiance is derived by taking the median signal for all pixels with center calibrated wavelength between 2312.7 to 2312.9 nanometers. Deep absorption features of $CH_4$ as well as $H_2O$ are avoided. Uncertainty

of the radiance measurements is propagated at each correction using gaussian error propagation. Note that the uncertainty on the radiance of a sounding is small and includes only the shot noise of the detector, photon noise and uncertainties of the calibration data included during L1b processing.

## 3.1    Lambertian surface and atmospheric correction assumptions

For our analysis, the surface is assumed to be Lambertian in nature. It is well documented that off-nadir soundings are affected

by non-Lambertian effects (e.g. Bruegge et al., 2019a). Although correction routines can be applied, these often rely on ground measurements to derive the BRDF correction factors within a modified Rahman-Pinty-Verstraete (mRPV) model (Rahman et al., 1993b, a). Even in such a well-characterized site such as the Railroad valley playa, differences larger than 5% are seen in normalization factor (Bruegge et al., 2019b, a). These differences appear to be dependent on measurements used to derive the free parameters in a mRPV model. Ground measurements are not available for any of the sites listed in Table 1. Measurements

from space, such as MODIS or MISR, are typically of insufficient quality to reliably improve the data due to saturation effects and/or lack of wavelength coverage (Schaaf et al., 2011; Bruegge et al., 2019a). This is particularly apparent at larger viewing

zenith angles. Various studies Bruegge et al. (2019b, a); Kuze et al. (2014)indicate that this assumption will cause the largest scatter. In theory, the very large swath of TROPOMI is well-suited to characterize this effect in more detail, following e.g. the results as presented in Bruegge et al. (2019b, a). However, this effect is considered to be beyond the scope of this work; exploration of BRDF effects can only be carried out at instrumented sites such as Railroad Valley due to the need for validation.

Similarly, the lack of atmospheric correction applied to the data also may introduce additional scatter and or effects. The most important effect is due to varying aerosol densities. This effect is assumed to be relatively small at 2.3 micron wavelengths. Analysis did reveal that these assumptions are the most likely origin for a secondary yearly variation introduced due to the dependency of solar angles to the date. This is revealed as a sine wave with a 365-day period. This sine wave is fitted and removed from the data. The effects are discussed in section 4.2. Future work (van Kempen et al., in prep) will explore non-Lambertian and atmospheric correction effects for TROPOMI SWIR wavelengths.

## 3.2 TROPOMI data quality

The TROPOMI-SWIR response has been shown to be extremely stable over the first year (van Kempen et al., 2019) with no detector degradation seen. This has been corroborated by the ongoing monitoring of the calibration carried out at SRON and KNMI[5] Degradation of the transmission is below 0.2% for the detector median. Regions of the detector also do not show degradation below the accuracy limits of the stability of the white light source. or the dedicated LED. This allows two key assumptions to be made for this work. First, change in signal above 2% is difficult to attribute to the instrument. Second, any degradation seen in the signal above the desert Sahara will be representative over a large part of the columns[6] associated with 2312.7 to 2312.9 nm, only excluding the top and bottom parts of the detector which correspond to viewing angles larger than 50 degrees. Solar calibration over this region shows deviations less than 0.2% over the time period presented in this paper.

## 4 Results

Table 3 shows the median radiance and its standard deviation (both in absolute and relative scales) of accepted soundings for all sites from a period starting April 28th 2018 to 1st October 2020. In addition to these statistical properties, the slope of a linear fit over 1,000 days is given. This linear fit can be used as a quantification of potential degradation. This can be compared to the SWIR stability as seen in van Kempen et al. (2019); Ludewig et al. (2020). E.g., this stability should be similar to the trends seen in the DLED, WLS and transmission, as shown in Figures 23, 24 and 25 of van Kempen et al. (2019). Note that the DLED has its own degradation. Figure 2 shows the complete time series of four sites of our sample. These four sites were chosen at random to represent the sample. The results of all other sites are presented in given in supplemental material. Figure 3 gives two examples of the sine wave that was fitted and subsequently corrected for.

Figure 4 shows the distribution of standard deviations and slopes of the linear fits of the full sample.

---

[5]See https://www.sron.nl/tropomi-swir-monitoring/ and http://mps.tropomi.eu/dashboard for daily updates and weekly and monthly reports.

[6]In following with agreed TROPOMI convention, the directions on the SWIR detector are defined as the spectral axis being the x-axis and the spatial (i.e. swath) direction being the y-axis.

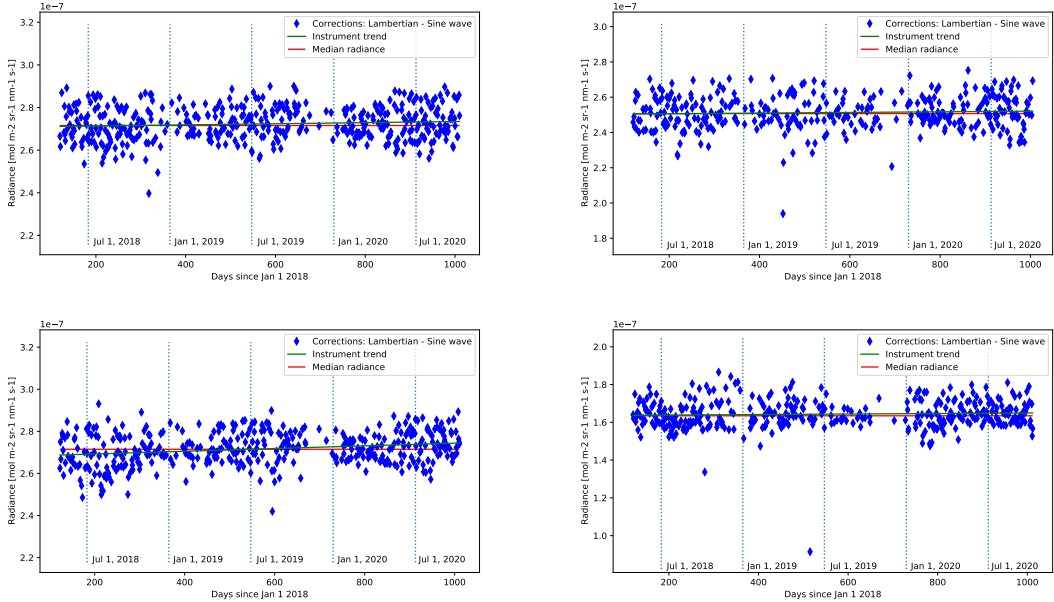

**Figure 2.** Continuum radiance of Egypt1 (top left), Algeria3 (top right), Libya3 (bottom left) and NamibiaPICSAND1 (bottom right) using a zenith angle filter of 50 degrees. Shown are all soundings over a period of more than 1,000 days, starting 28th of April 2018. The median radiance (red) and a linear fit (green) are shown with lines. Data has been corrected for the solar zenith angle at each individual overpass and assuming the surface is Lambertian. A yearly variation has been corrected using a fitted sine wave with a period of 365 days.

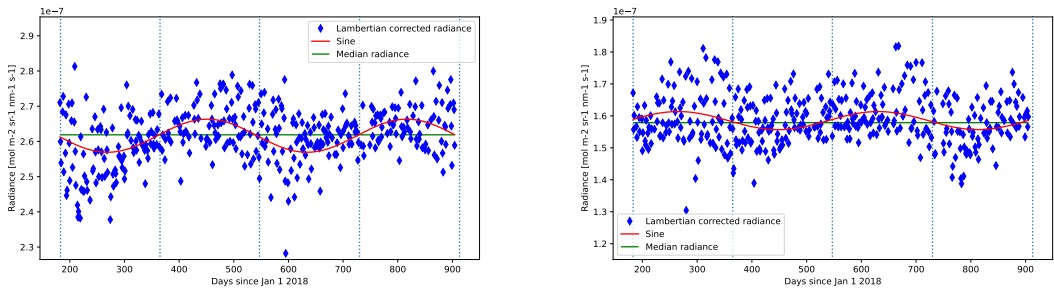

**Figure 3.** Continuum radiance of Libya3 (left) and NamibiaPICSAND1 ( right) using a zenith angle filter of 50 degrees without the yearly sine variation removed. Shown are all soundings over a period of more than 1,000 days, starting 28th of April 2018. The median radiance (green) and a fitted sine (red) are shown. Data has been corrected for the solar zenith angle at each individual overpass and assuming the surface is Lambertian. Data of late 2019 is included to better illustrate fit. During this time reliable cloud information has been sparse.

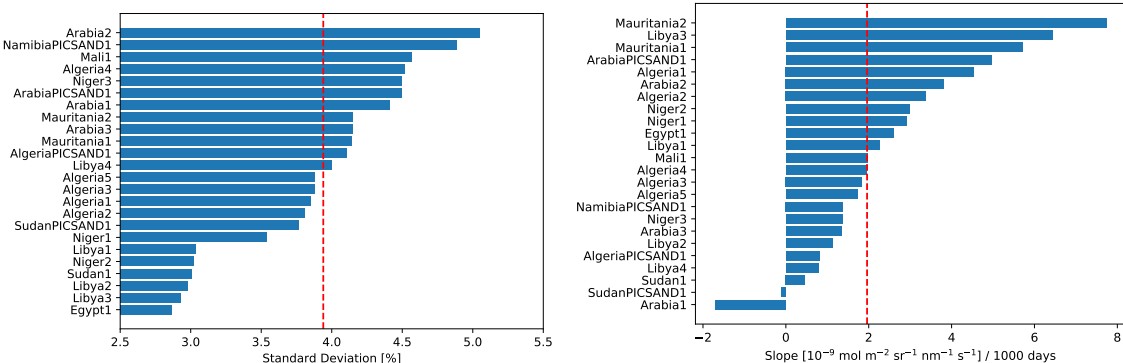

**Figure 4.** Sorted bar diagrams for the standard deviation of the radiance distribution of each site (left) and the slope of a linear fit, expressed in the slope per 1000 days (right). In red we show the median standard deviation and median slope for the distribution.

Interestingly, the standard deviation found for most sites is a) much larger than the uncertainty per measurement (which is estimated to be less then 0.1% Kleipool et al. (2018); van Kempen et al. (2019); Ludewig et al. (2020)), b) relatively constant for all sites in both time and magnitude. A spread of 8 to $11 \times 10^{-9}$ mol m$^{-2}$ sr$^{-1}$ nm$^{-1}$ s$^{-1}$, equivalent to 3-5% is found for all sites.

The results of van Kempen et al. (2019) show that instrumental variation in measurement to measurement due to detector noise is much smaller, and that the uncertainty from instrumental effects is dominated by photon noise only, which is estimated to be on the order of 0.1%). The higher measurement to measurement variation reported in Ludewig et al. (2020) seen in the irradiance were attributed to electronic noise. However, due to more in-depth analysis carried out, it is now attributed to surface roughness of the diffusers (van Kempen & Ludewig, priv. comm). The distribution of viewing angles from TROPOMI SWIR

over a full year is large, due to the large swath of TROPOMI that allows for daily global coverage.

From investigations of the TROPOMI-SWIR data above the RailRoad Valley site compared to on-ground measurements, we attribute the derived standard deviation to non-Lambertian reflections of the sandy surface (van Kempen et al., in prep).

Individual site heterogeneity is considered to be a small effect. Site heterogeneity would effect the results due to the varying size of the TROPOMI pixel, and exhibit itself as a pattern repeated every 16 days, equivalent to the orbital cycle of S-5p. This

pattern varies by site and depends on the more precise BRDF proporties of the surfaces. No clear evidence for patterns on these timescales can be seen or derived from Figure 2. Heterogeneity would also be detectable as a correlation between the values SHOM of 20 km or 100 km of Table 2 of Bacour et al. (2019) and the derived standard deviation values of Table 3. Figure 5 shows the lack of correlation between the given site homogeneity at scales of 20 km and the standard deviation of the time series given in Table 3.

Linear fits found are very shallow, with typical number between 0 and $5 \times 10^{-9}$ mol m$^{-2}$ sr$^{-1}$ nm$^{-1}$ s$^{-1}$ per 1,000 days. This equates to a maximum increase in transmission of 0.7% per year. The average of the slope is equal to 0.3 % per year. Note

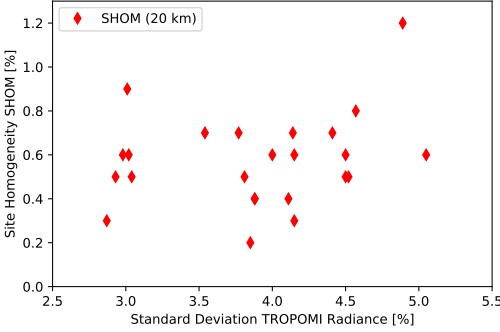

**Figure 5.** Comparison of the relative standard deviation of the time series of radiance with the site homogeneity at 20 km (SHOM) given in Bacour et al. (2019), revealing a lack of correlation.

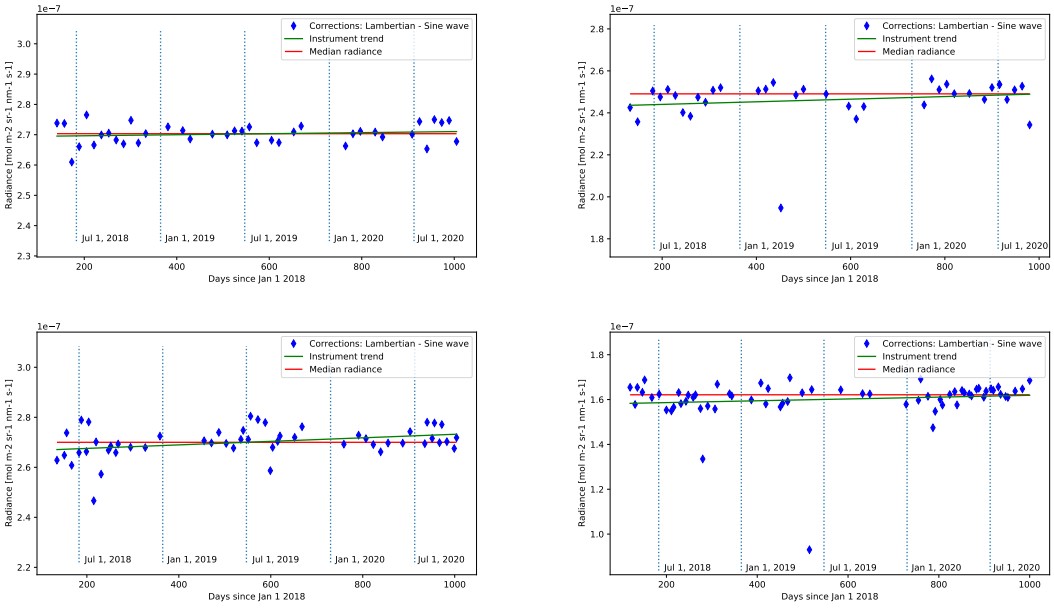

**Figure 6.** Continuum radiance of Egypt1 (top left), Algeria3 (top right), Libya3 (bottom left) and NamibiaPICSAND1 (bottom right) using a zenith angle filter of 7.5 degrees. Shown are all soundings over a period of more than 1,000 days, starting 28th of April 2018. The median radiance (red) and a linear fit (green) are shown with lines. Data has been corrected for the solar zenith angle at each individual overpass and assuming the surface is Lambertian. A yearly variation has been corrected using a fitted sine wave with a period of 365 days. Other sites are given in supplementary material.

that many sites are found at lower values, and a few even showing a negative slope. Given the typical standard deviation, we attribute the slopes to statistical variations. The impact on SWIR instrument degradation is discussed below.

**Table 3.** Median radiance, standard deviation (both absolute and relative) and slope of a linear fit for the full sample.

| Site | Median Radiance $[10^{-7}]$ $\mathrm{mol\ m^{-2}\ sr^{-1}\ nm^{-1}\ s^{-1}}$ | St. Dev $[10^{-9}]$ | % | Slope $[10^{-9}]$ $\mathrm{mol\ m^{-2}\ sr^{-1}\ nm^{-1}\ s^{-1}}$ per 1000 days |
|---|---|---|---|---|
| Algeria1 | 2.44 | 9.4 | 3.85 | 4.54 |
| Algeria2 | 2.51 | 9.6 | 3.81 | 3.38 |
| Algeria3 | 2.51 | 9.7 | 3.88 | 1.84 |
| Algeria4 | 2.57 | 11.6 | 4.52 | 1.95 |
| Algeria5 | 2.56 | 9.9 | 3.88 | 1.73 |
| AlgeriaPICSAND1 | 2.54 | 10.4 | 4.11 | 0.82 |
| Arabia1 | 2.38 | 10.5 | 4.41 | -1.71 |
| Arabia2 | 2.31 | 11.7 | 5.05 | 3.82 |
| Arabia3 | 2.41 | 10.0 | 4.15 | 1.35 |
| ArabiaPICSAND1 | 2.45 | 11.0 | 4.50 | 4.96 |
| Egypt1 | 2.72 | 7.9 | 2.87 | 2.59 |
| Libya1 | 2.76 | 8.4 | 3.04 | 2.26 |
| Libya2 | 2.74 | 8.2 | 2.98 | 1.12 |
| Libya3 | 2.71 | 8.0 | 2.93 | 6.44 |
| Libya4 | 2.65 | 10.6 | 4.00 | 0.78 |
| Mali1 | 2.55 | 11.6 | 4.57 | 1.97 |
| Mauritania1 | 2.63 | 10.8 | 4.14 | 5.71 |
| Mauritania2 | 2.63 | 10.9 | 4.15 | 7.73 |
| NamibiaPICSAND1 | 1.63 | 8.0 | 4.89 | 1.38 |
| Niger1 | 2.75 | 9.7 | 3.54 | 2.91 |
| Niger2 | 2.49 | 7.5 | 3.02 | 2.99 |
| Niger3 | 2.36 | 10.6 | 4.50 | 1.36 |
| Sudan1 | 2.59 | 7.8 | 3.01 | 0.46 |
| SudanPICSAND1 | 2.66 | 10.1 | 3.77 | -0.10 |

## 4.1 Nadir view

Given the influence of non-lambertian reflection on the standard deviation, the stability of the four sites with various quality (Egypt1, Algeria3, Libya3 and NamibiaPICSAND1, see Fig. 6) is shown using only viewing angles less than 7.5 degrees from nadir. The results are given in Table 4 with the full results given for completeness. Differences are marginal, with the largest differences seen in the linear fit. For some, the standard deviation improved. The main difference was found in the reliability of the removal of the yearly variation through the 365-period sine wave. Interestingly, the fit parameters were nearly

**Table 4.** Median radiance, standard deviation (both absolute and relative) for the sample with a Instrument Zenith Angle (IZA) filter of 7.5 degrees.

| Site | IZA 7.5 degrees | | |
|---|---|---|---|
| | Median Radiance | St. Dev | |
| | $10^{-7}$ | $10^{-9}$ | % |
| | mol m$^{-2}$ sr$^{-1}$ nm$^{-1}$ s$^{-1}$ | | |
| Algeria1 | 2.44 | 8.1 | 3.3 |
| Algeria2 | 2.51 | 7.4 | 3.0 |
| Algeria3 | 2.49 | 10.2 | 4.2 |
| Algeria4 | 2.58 | 6.3 | 2.4 |
| Algeria5 | 2.56 | 6.2 | 2.4 |
| AlgeriaPICSAND1 | 2.52 | 10.0 | 4.0 |
| Arabia1 | 2.39 | 7.2 | 3.0 |
| Arabia2 | 2.30 | 7.7 | 3.3 |
| Arabia3 | 2.42 | 6.5 | 2.7 |
| ArabiaPICSAND1 | 2.45 | 8.3 | 3.4 |
| Egypt1 | 2.70 | 3.2 | 1.2 |
| Libya1 | 2.75 | 7.2 | 2.6 |
| Libya2 | 2.76 | 6.2 | 2.2 |
| Libya3 | 2.70 | 5.9 | 2.2 |
| Libya4 | 2.63 | 8.7 | 3.3 |
| Mali1 | 2.50 | 11.0 | 4.4 |
| Mauritania1 | 2.63 | 12.4 | 4.8 |
| Mauritania2 | 2.53 | 12.3 | 4.9 |
| NamibiaPICSAND1 | 1.62 | 9.9 | 6.2 |
| Niger1 | 2.72 | 12.2 | 4.5 |
| Niger2 | 2.48 | 5.02 | 2.0 |
| Niger3 | 2.36 | 15.0 | 6.5 |
| Sudan1 | 2.59 | 7.3 | 2.8 |
| SudanPICSAND1 | 2.69 | 6.4 | 2.4 |

identical as that derived above, with only the reliability of this fit found to be significantly worse. This is easily explained due to the significantly lower amount of data used as input for the fit. This also corroborates the conclusions on the influence of heterogeneity. Any effects of site heterogeneity would be nearly eliminated as the pixels are nearly identical in size and viewing the same area. This would result in a severely reduced standard deviation, which is not observed in Table 4.

**Table 5.** Amplitude and period offset of fitted and subtracted sine waves.

| Site | Amplitude $10^{-9}$ [mol m$^{-2}$ sr$^{-1}$ nm$^{-1}$ s$^{-1}$] | Offset Days |
|---|---|---|
| Algeria1 | 6.86 | -14.9 |
| Algeria2 | 7.12 | -15.5 |
| Algeria3 | 4.99 | 2.4 |
| Algeria4 | 4.03 | 24.9 |
| Algeria5 | 4.70 | 18.6 |
| AlgeriaPICSAND1 | 6.43 | 0.3 |
| Arabia1 | 1.71 | 203.5 |
| Arabia2 | 0.98 | 35.4 |
| Arabia3 | 4.61 | 84.3 |
| ArabiaPICSAND1 | 4.67 | 79.6 |
| Egypt1 | 4.51 | 44.6 |
| Libya1 | 3.94 | -7.3 |
| Libya2 | 4.96 | -8.8 |
| Libya3 | 3.58 | -12.9 |
| Libya4 | 5.99 | 32.2 |
| Mali1 | 11.57 | -41.4 |
| Mauritania1 | 11.35 | -40.5 |
| Mauritania2 | 9.95 | -42.0 |
| NamibiaPICSAND1 | 3.14 | 133.8 |
| Niger1 | 6.03 | -29.1 |
| Niger2 | 4.21 | -22.7 |
| Niger3 | 4.47 | -29.0 |
| Sudan1 | 5.21 | 1.38 |
| SudanPICSAND1 | 7.85 | -16.6 |

## 4.2  BRDF

A clear limitation of our analysis is the assumption of a Lambertian surface over the desert sites, as detailed in section 3.1. A more thorough analysis using a correct reflectance model over each desert surface should be able to improve the model. This is typically done over instrumented sites such as RailRoad Valley. As none of the sites of Table 1 contain ground instrumentation that covers the 2.3 micron band, several alternatives were explored to achieve this. However, these were unsuccesfull:

- Over these desert sites, the MODIS BRDF product MCD43A1 was found not to be accurate enough. Due to saturation over the bright desert sites, in particular at longer wavelengths, the product includes significant gaps in the data coverage. In addition, any accepted product shows large uncertainties.

- The VIIRS BRDF dataproduct VNP43DNBA1 is not saturated, but appears to lack the so-called 'hot spot' in key directions, introducing a systematic error.

- Over instrumented sites such , the accuracy of the correction of BRDF effects appears to be $5\%$ (Bruegge et al., 2019a/2019b, Kuze et al., 2014). A $5\%$ uncertainty is currently larger than the $3\%$ spread/variation we see.

These conclusions corroborate the conclusion that for implementation of BRDF corrections of TROPOMI measurements, instrumented sites play an important role. In addition, one needs to explore differences in site composition at 2.3 micron for it to be applicable to Saharan desert sites.

## 4.3 Origin of yearly variation

The sine wave used to correct an observed yearly variation is attributed to the BRDF effect It is not instrumental in nature given the results of van Kempen et al. (2019) and ongoing monitoring (see www.sron.nl/tropomi-swir-monitoring/) and should thus be attributed to a feature present in the surface. The amplitudes and period offsets with respect to January 1st 2018 are given in Table 5. Example fits are given in the appendix. It appears that variation in this yearly variation is regionally distributed. Sites in the south-western variation (Mauritania, Mali, Niger and Algeria1 and Algeria2, have a larger amplitude (6 to 11) period offset with a small negative offset in days w.r.t. to the reference day of Jan 1 2018 while eastern Sahara sites (Libya, Egypt, Sudan) tend to have smaller amplitudes (4-5) and show a slight positive offset. The Namibian desert site is clearly different due to its location on the southern hemisphere. The sites in the Arabian desert are grouped south (Arabia1 and Arabia2) and north (Arabia3 and ArabiaPICSAND1) in amplitude, but show no correlation in period. Arabia1 has a relatively large offset.

As to the origin, four possibilities are entertained. Several others are either disqualified or limited to relaively small contributions. The first is shadowing of dune formation, caused by site heterogeneity. Bacour et al. (2019) investigated site homogeneity for both the original positions of Cosnefroy et al. (1996) and their own more optimal locations. The latter, used here, are significantly more homogeneous at scales of both 20 and 100 km. Given the typical pixel size of TROPOMI, homogeneity at 20 km is relevant . Differences are seen in Google maps (Bacour et al., 2019), but in general the sites are homogeneous down to a few percent or less. If dune formation would be a large factor, minima would be expected around mid-December due to the solar zenith angle. This minimum corresponds to an offset in days equal to approximately 80. In addition, the sole southern hemisphere location would show a minimum during late June and an offset of 260 days. This offset would also not be dependent on site, while amplitude can be. Both are not seen Table 3. However, dune formation and site heterogeneity in general can be more chaotic and thus needs to be investigated further.

The second possibiliy is a dependency to rock or sand composition for reflectance and BRDF, another component in site heterogeneity. Our data is insufficient to make any conclusions on this. However, it would be related to location and size of

hotspot and would be incompatible with the observation that the yearly variation is detected with similar amplitudes using only nadir overpasses. This hypothesis does warrant deeper investigation.

Third, given the ISRF width (van Hees et al., 2018) some wings of absorption features are within the range. Such line absorption could show a yearly variation.These absorptions are very small ($<0.5\%$) and are not part of either $CH_4$ or $H_2O$ absorption features. Identifying any line contamination beyond $CH_4$ and $H_2O$ is considered beyond the scope of this paper. Given the expected magnitude, this possibility is considered to be only a very remote possibility.

Last but not least, the lack of atmospheric correction likely influences the results. At SWIR continuum wavelengths this correction is dominated by varying quantities of aerosol in the atmosphere as a function of atmosphere height. This effect likely can contribute to yearly variations, but due to the opacity of aerosols at these wavelengths (0.05 or less) is expected to be smaller than the BRDF effects.

Several possibilities are ruled out. First, any instrument stability that would be yearly in scope would be detectable either with the on-board calibration and/or reflectance average that is also monitored (van Kempen et al., 2019; Ludewig et al., 2020). These show limits less than 0.3%, in line with the results of the linear fits. Another possibility is the varying distance between the Sun and the Earth. Although this variation is expected to be of similar magnitude ($\sim2.25\%$, it would also cause two features not observed. First perihelion occurs near January 5th. Sine offsets would thus be consistly fitted at 90 or +270 days in Table 5. Deviations from this fit are not expected to be larger than 10%. However fits with a complete 180 degree phase difference are found. TROPOMI radiance is corrected for the Earth-Sun distance (See TROPOMI L1b ATBD: S5P-KNMI-L01B-0009-SD).

Although the second option (BRDF) appears to be the most likely, the other three options cannot be excluded, in particular the effect of aerosol in the atmospheric correction, nor can it be excluded that all contribute to the variation and/or spreads observed. A future publication (van Kempen et al., in prep) will investigate the BRDF and the effect of the atmospheric corrections including aerosols, including its yearly variation. This future publication will carry out the analysis over a single instrumented site (RailRoad Valley) to be able to accurately provide validation against ground measurements.

## 5 Conclusions

The results above show that the sample of Saharan PICS are well-suited as a validation of the instrument stability, both using viewing angles up to 50 degrees and to only 7.5 degrees. The standard deviations seen are similar from site to site. This was achieved by a strict filter on clouds (cloud coverage of $<0.02$) and correcting for the solar zenith angle during any overpass. In addition, observed yearly variations were removed by fitting a sine wave with a period of a single year. The results clearly show no evidence for TROPOMI-SWIR instrument degradation or increased transmission. The results validate the conclusions shown in van Kempen et al. (2019) with a limit of 0.5% over the first 1,000 days. Note that the internal calibration suite shows no degradation or increased transmission down to levels of $\sim0.1\%$ over the first year. Subsequent monitoring[7] has revealed this to hold over the first 1,000 days as well.

---

[7]See www.sron.nl/tropomi-swir-monitoring/

The lack of difference using filters with both 50 and 7.5 degrees is somewhat surprising given the attributed influence of non-lambertian reflectances (which peak at angles of 20-30 degrees from zenith) and the inclusion of larger TROPOMI pixels (due to the swath size of TROPOMI). Note however that following Bacour et al. (2019) most sites were assumed to be spatially homogeneous. This likely explains the lack of influence of larger pixels at larger viewing angles. Subsequently, we can conclude that the spatial homogeneity conclusions indeed apply to SWIR wavelengths following this study.

The variation from site to site is also relatively small, but do appear to be correlated by region. The Namibian desert has a significantly lower radiance, while the sites in the Arabian desert only show a lower radiance by about 3-5% as compared to the Saharan Desert. Differences within the Sahara (i.e. east-west or per country) appear to be marginal and warrant further investigation.

The assumption of a Lambertian surface limits the accuracy of the data, and provides a limit to the interpretation of the data. It very likely is the origin of the observed spread in each time series, as well as either the main contributor or a significant part of observed systematics variations such as the observed yearly variation. TROPOMI is an ideal instrument to explore BRDF effects out to very large viewing angles. However, given the limitations of space-based BRDF products as discussed in section 4.2, conclusions given there, and recent conclusions of the BRDF studies in e.g. RailRoad Valley (Bruegge et al., 2019a), such studies must include validation by usage of instrumented sites.

In conclusions, future SWIR spectrometers should, if possible, prefer the usage of internal calibration sources or solar calibration over the usage of PICS as the primary means for instrument calibration. However, for so-called small-sat applications with total weight and/or life-limited instrument constraints, results presented above show that monitoring of suitable PICS offer a suitable, albeit a relatively more inaccurate, alternative. Note however that at high resolution, several limitations introduced by heterogeneity will be less relevant. Currently, the accuracy of the correction of BRDF effects is the greatest limitation, which should be explored over instrumented sites.

*Code and data availability.* All L1b-data is freely available through the Copernicus Open Access hub (https://scihub.copernicus.eu). The pys5p code that is the basis of the analysis code can be found on Zenodo (https://zenodo.org/record/4073256) under DOI 10.5281/zenodo.4073256 intermediate results are available through SRON.

*Competing interests.* There are no known competing interests.

*Author contributions.* The work was started by FO under supervision of TvK as part of an internship. TvK expanded conclusions and additional work, including writing the paper. The base software packages were written and provided by RvH.

*Acknowledgements.* We thank the team that has realized the TROPOMI instrument, consisting of the partnership between Airbus Defence and Space Netherlands, Royal Netherlands Meteorological Institute (KNMI), SRON Netherlands Institute for Space Research (TNO), and Netherlands Organisation for Applied Scientific Research (TNO), Netherlands Space Office (NSO), and European Space Agency. This research contains modified Copernicus Sentinel data of 2018, 2019 and 2020. The work of TvK is funded by the TROPOMI national program from the NSO. We would like to acknowledge the fruitful discussions with several members of the Earth Science Group at SRON (Ilse Aben, Paul Tol, Tobias Borsdorff, Jochen Landgraf and Alba Llorente).

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
