# Peer review of "Monitoring the TROPOMI-SWIR module instrument stability using desert sites"

_Atmospheric Measurement Techniques, 2020_

## Referee Comment (RC1)

**Monitoring the TROPOMI-SWIR module instrument stability using Desert sites**

Tim A. van Kempen, Filippo Oggionni2 and Richard M. van Hees

**General comments**

The proposed publication provides a new contribution to typical key topics for space instrumentation: the monitoring of any trend in the instrument response, and proposed solutions to compensate the degradation (trend corrections). For space payloads designed for radiance/irradiance measurements, it is important to maintain the radiometric scales and instrument response. It is here presented for the SWIR channel of TROPOMI. Usually, the strategy is to use internal light sources or external references. This publication wants to demonstrate that PICS (here mainly selected in Arabia and North Africa) can be used for the SWIR channel and L1b data. The paper is clearly written. It presents the different tools that must be applied for the monitoring of instrumental trends. There is good quality for the discussion and the argumentation, for the selection of sites, the constraints and filtering of the data, the data processing and associated uncertainty calculations and conclusions. The proposed paper is of good quality and this work is clearly performed by a team expert in their own instrumentation and their general scientific domain (atmospheric sciences, monitoring of trace gases).

**Specific comments**

It is well understood that a main discussion concerns the standard deviation of the PICS radiance monitoring (in the range of 3-5%), larger than 'expected', which cannot be explained by instrumental uncertainty calculation for single measurement (~0.1%, line 129). The discussion is conducted on possible non-lambertian effects, but inconclusive as wrote in the conclusions (line 179) due to the lack of difference using filters '50°' and '7.5°', but it is not a main issue for the publication to present open questions. However, these uncertainties of 3-5% impacts also the quality of instrument trend retrieval using PICS in the SWIR, as clearly seen in the Figure 3 right (large distribution of slopes). So, one crucial objective of this paper (to validate the use of PICS in SWIR in addition to internal lamps of TROPOMI) is maybe not enough discussed: only in the conclusions where the average value (0.3%/year) obtained from PICS monitoring is compared to the results of internal calibration (line 177). Maybe some more sentences on the opinion of the researchers could be added. For example, to answer to these questions: better to use internal lamps? better to use PICS? important to use both for cross-checked, while knowing that PICS monitoring will not provide a lower uncertainty than the use of internal light sources?

- Concerning the yearly variation correction: this topic is well discussed even if it is still an open question. It is clearly well corrected (sine correction), but it is to note that the amplitude of these events is not so negligible (in the 10-9 radiance unit, the same order than the standard uncertainty of the final, filtered products).

**Technical corrections**

- Line 72 '....to track the temporal variability of the SWIR and NIR channel continuum signal ...'. Why to invoke here the NIR spectral range? It is not discussed elsewhere in the publication.
- Maybe, some sentences or paragraph structure could be readjusted to present a better logical link between the filters (viewing angle, cloud cover, irradiance measurement, overpass separation, instrument zenith angle, ...) and the discussion on them. For example, the applied filters are described at the beginning of section 3. One of them (' ... the viewing zenith angle must be smaller than 50 degrees ...') is presented at line 82. Then a discussion tarts at line 88 on the pixel size, and the need to maintain any side length below 20 km (to have a pixel located at less than 20 km away from a PICS), which justify the need to reduce the viewing angle below 50° (line 94). So the explanation comes here. Also at line 82 ('... a successful and valid irradiance measurement should have been taken within a day ...'). It seems redundant with the sentences at line 111 ('... Last but not least, soundings are required to have a valid irradiance measurement using TROPOMI-SWIR within a day ...') but in fact, the discussion of this filter comes here.
- It could be useful also to add the column 'slope' in Table 4. The reader can find the new slopes (after a filtering of zenith angle to 7.5°), but it is not possible to compare the numerical values presented in Figure 3, even if you wrote some sentences for the discussion (line 147-148-149: Interestingly, the fit parameters were nearly identical ...').

---

## Author Comment (AC1)

*Referee 1*

**General comments**

The proposed publication provides a new contribution to typical key topics for space instrumentation: the monitoring of any trend in the instrument response, and proposed solutions to compensate the degradation (trend corrections). For space payloads designed for radiance/irradiance measurements, it is important to maintain the radiometric scales and instrument response. It is here presented for the SWIR channel of TROPOMI. Usually, the strategy is to use internal light sources or external references. This publication wants to demonstrate that PICS (here mainly selected in Arabia and North Africa) can be used for the SWIR channel and L1b data. The paper is clearly written. It presents the different tools that must be applied for the monitoring of instrumental trends. There is good quality for the discussion and the argumentation, for the selection of sites, the constraints and filtering of the data, the data processing and associated uncertainty calculations and conclusions. The proposed paper is of good quality and this work is clearly performed by a team expert in their own instrumentation and their general scientific domain (atmospheric sciences, monitoring of trace gases).

➢ Dear Referee, thank you fort he kind comments

**Specific comments**

- It is well understood that a main discussion concerns the standard deviation of the PICS radiance monitoring (in the range of 3-5%), larger than 'expected', which cannot be explained by instrumental uncertainty calculation for single measurement (~0.1%, line 129). The discussion is conducted on possible non-lambertian effects, but inconclusive as wrote in the conclusions (line 179) due to the lack of difference using filters '50°' and '7.5°', but it is not a main issue for the publication to present open questions. However, these uncertainties of 3-5% impacts also the quality of instrument trend retrieval using PICS in the SWIR, as clearly seen in the Figure 3 right (large distribution of slopes). So, one crucial objective of this paper (to validate the use of PICS in SWIR in addition to internal lamps of TROPOMI) is maybe not enough discussed: only in the conclusions where the average value (0.3%/year) obtained from PICS monitoring is compared to the results of internal calibration (line 177). Maybe some more sentences on the opinion of the researchers could be added. For example, to answer to these questions: better to use internal lamps? better to use PICS? important to use both for cross-checked, while knowing that PICS monitoring will not provide a lower uncertainty than the use of internal light sources?

> To better illustrate these points, several lines have been added to the conclusions. The main conclusion is that even the average value does not constrain TROPOMI degradation very well. For that one needs to let go of the Lambertian assumption. However, this point is not stressed too importantly due to the limitations of the method here. Current research in our group is underway to understand the BRDF effects over RailRoad Valley. However, it is clear that these effects are complicated and currently cannot be verified over non-instrumented PICS sites.

- Concerning the yearly variation correction: this topic is well discussed even if it is still an open question. It is clearly well corrected (sine correction), but it is to note that the amplitude of these events is not so negligible (in the 10-9 radiance unit, the same order than the standard uncertainty of the final, filtered products).

> Very likely this is also a BRDF effect due to the position of the sun.

**Technical corrections**

- • Line 72 – '....to track the temporal variability of the SWIR and NIR channel continuum signal ...'. *Why to invoke here the NIR spectral range? It is not discussed elsewhere in the publication.*

➢ Corrected, this was a mistake from an earlier version.

- • Maybe, some sentences or paragraph structure could be readjusted to present a better logical link between the filters (viewing angle, cloud cover, irradiance measurement, overpass separation, instrument zenith angle, ...) and the discussion on them.
  For example, the applied filters are described at the beginning of section 3. One of them (' ...the viewing zenith angle must be smaller than 50 degrees ...') is presented at line 82. Then a discussion tarts at line 88 on the pixel size, and the need to maintain any side length below 20 km (to have a pixel located at less than 20 km away from a PICS), which justify the need to reduce the viewing angle below 50° (line 94). So the explanation comes here. Also at line 82 ('... a successful and valid irradiance measurement should have been taken within a day ...'). It seems redundant with the sentences at line 111 ('... Last but not least, soundings are required to have a valid irradiance measurement using TROPOMI-SWIR within a day ...') but in fact, the discussion of this filter comes here.

➢ The structure of this section was reconsidered.

- • It could be useful also to add the column 'slope' in Table 4. The reader can find the new slopes (after a filtering of zenith angle to 7.5°), but it is not possible to compare the numerical values presented in Figure 3, even if you wrote some sentences for the discussion (line 147-148-149: Interestingly, the fit parameters were nearly identical ...').

➢ This was omitted on purpose to avoid over-interpretation. If done correctly, it would also need to include the errors on the slopes in both tables. These errors are (due to BRDF and yearly nonsinusoidal variations in a few plots, see append) not true standard deviations. As such only the sentence was included.

---

## Author Comment (AC2)

**Referee 2**

The manuscript Monitoring the TROPOMI-SWIR module instrument stability using desert sites by van Kempen et al outlines an approach for TROPOMI SWIR instrument stability monitoring using PICS site over deserts. While the manuscript is in general well written, I do have some high level questions/concerns regarding the ultimate utility of the study for TROPOMI validation apart from stating that it is in line with the onboard calibration routines. In the following, I will briefly describe some of these concerns, followed by a few detailed comments on typose, etc.

Dear referee, thank you for your constructive criticism. They proved useful. With the replies below, we hope to address your concerns.

with kind regards,

Tim

**Major comments:**

As far as I can see at the moment, the entire outcome of this investigation is a bulk characterization of the 2D detector response at one specific small wavelength band averaged over the entire spatial domain. However, I haven't seen any discussion on the implication of this limiting factor at all, which I consider rather large. First of all, the across-track pixels might all degrade to a different degree or have an overall different absolute calibration error.

We know from the monitoring of the TROPOMI-SWIR detector that these hypotheses are not applicable and that the detector is very stable. This is (in part) reported in van Kempen et al., 2019 for the first year of operations. Monitoring of the calibration can be found on the website of SRON: https://www.sron.nl/tropomi-swir-monitoring as well as derived from the weekly reports posted on http://mps.tropomi.eu/reporting corroborate these results for the full mission so far.

The referee is correct however that these points are far too implicitly assumed in the paper. We have added a sub-section in Section 3 to make these conclusions explicit. However, their response is also impacted by potential BRDF effects as each across track element has its own viewing zenith angle. Given the data density of TROPOMI, I was somewhat surprised that the authors didn't try to at least disentangle some of the across-track element variations.

We started with the analysis of potential BRDF effects, but following results of e.g. Bruegge et al., 2019 and a collaboration with the JPL group using RailRoad Valley data, we quickly concluded that this is a much more complicated analysis with a significant amount of relevant parameters (BRDF of the surface, overpass time and angles of both sun and instrument). We are currently investigating these effects in much higher detail over RRV with the GOSAT and OCO teams. But to apply these correctly to the results in this work clearly was beyond the scope of this paper.

This 'problem' was only succinctly described in lines 100-105 in the preprint version, and I agree with the referee it was far too concise. This has been expanded to a new subsection and point to the future publication on this topic.

At the moment, I am somewhat uncertain what new information this manuscript reveals, especially as the scatter is surprisingly and the variations in slopes quite variable too. The authors would have to better explain the added value of this method (on top of the on-board calibration, which can characterize the entire FPA response).

The most relevant part of the information is the feasibility of this method for spectrometers in the SWIR region. I agree the scatter is surprising, and we are working on this to derive the methods to improve this (see above). The variations in slope are both an effect of the scatter and the surface properties themselves.

For TROPOMI itself, the added value on top of on-board calibration and a relatively poor validation, is indeed relatively marginal, although it would be the first L1b validation for the TROPOMI-SWIR. That in itself would be added value to the scientific community.

However, for future proposed small-sat missions where total weight is a major concern, the omission of on-board calibration will become more defendable given the existence and details of these methods. In particular, for SWIR missions this has not yet been demonstrated and is a major concern with some missions in development.

The origin of the scatter would have been an interesting feature to dig deeper into, but the authors chose not to, which is somewhat dissatisfying, as this could have been valuable to the community.

We agree with the referee, and are working on producing such a result in an upcoming publication (see above).

**Minor issues:**

Line 31: "calibrated column densities" These are retrieved products from calibrated spectra, not itself "calibrated" datasets (maybe validated and some post-hoc "calibration" like bias correction applied)

corrected

Line 48: allpart

corrected

Line 50++ Here you mention all kinds of impact factors but then chose to ignore all of them. Why not use TROPOMI and its large swath to actually see whether you can detect BRDF effects that can clearly be separated from detector effects across the spatial domain.

See comment above. We are carrying out a study to do this. This is now rephrased.

Line 57: "but also suffers from inaccuracies" Across the manuscript, statements like this are scattered. If you point out a weakness of an instrument, you will have to justify the statement with a citation or elaborate how you come to the conclusion. However, you can't just make a statement like this out of thin air in a peer-reviewed publication. Also, what does "most complete" mean in this context?

Rephrased

Line 57: "due to its very wide swath opening" --> just swath is fine, swath opening sounds awkward

Rephrased

Line 65 --> used for monitoring the stability of a large numberg of ...

corrected

Line 76: Why only every 5 days? The cloud cover should be low, so I dont understand the 5 day limit. Is it the overlap requirement? With MODIS data

being available, you should also be able to determine the impact of the exact spatial overlap across variable surfaces with slightly varying albedos. As far as I can see, no attempt has been made to compare against MODIS data (e.g. to look at sub-pixel variability, etc).

These refer to daily overpasses, and the occurrence, once every 5 days, of multiple overpasses within 1 day. Higher latitude sites (e.g. Gobi desert) would be covered even more often. This is unrelated to cloud cover, but originates with TROPOMI orbit parameters. In effect over 95% of the data is usable.

The referee is correct that no attempt was made to compare this to MODIS data. However, for a similar reason as the the BRDF, we quickly concluded that subpixel MODIS analysis rapidly increases the complexity of this comparison. In addition saturation effects of MODIS data due to the typical Solar zenith angles in combination with the high reflectance of desert surface limits this analysis severely.

**Table 1 caption typos: cooerdinates... variatility...**

corrected

**Line 86++ Please better explain the cloud screening, at least give a proper citation. Has any screening for desert dust events been performed? Any other filters?**

This section was re-ordered. Screening for desert dust events have not been performed.

**Line 94: Why did you choose 50 degrees as cutoff even though this basically omits a non-significant fraction of TROPOMI's FPA? Have you checked whether adding the few additional degrees make any differences? Did you consider separating out the FPA (and thus VZA) dependence as mentioned above?**

The choice of 50 is motivated by an estimate. We wanted as much data as possible without being heavily affected by the Lambertian assumption. Although the angle dependency is already apparent at smaller angles due to e.g. the hotspot (Bruegge et al., 2019), the data beyond 55 degrees suggests that BRDF effects of the SWIR even deviate from the angle-dependent BRDF effects as modelled by Bruegge et al., 2019. Without understanding both the range of BRDF effects themselves, as well as any deviations from their angle-dependency as observed compared to wavelengths, it was estimated as too impactful at large angles. Thus, the Lambertian assumption was made.

There isn't a lot of difference up to an angle of 52, but significant higher scatter when including the full FPA or any angle from 55 degree or higher was seen.

Line 105: "are of insufficient quality to reliably improve the data" Please see my comment above. Without citation or justification with analysis, this statement is misplaced at beast and mean-spirited at worst. Any judgement statement like this requires corroboration.

**Rephrased**

**Line 106: "A choice was made" What was the rationale of that choice? Did you consider the tradeoffs? Why not bin the analysis by viewing angles and see whether the scatter is reduced?**

This binning scheme was calculated. The scatter was not reduced and appears to be dominated by BRDF effects depending on solar and instrument viewing angles. As such it was omitted.

**Line 112: affect -> affects**

**Corrected**

Line 116: "using standard mathematical rules" like what? Just gaussian error propagation? Please be specific if you can, esp. if it doesn't take up more space than "using standard mathematical rules", which is rather vague.

**Corrected**

Figure 2: Ths scatter is indeed large and clear outliers exists. Are these actually single measurements? If yes, can they be color-coded by the detector position or VZA (plus and minus)? Did you try to figure out why a few were low outliers by looking at the conditions during that time (or the specific detector position?)

We investigated this by color-coding, but found it to be a 5-variable problem (detector position, instrument zenith, instrument azimuth, solar azimuth, solar zenith). Although detector position and viewing zenith are related, small changes in viewing zenith show up as a few rows difference on the detector and thus give rise to larger changes.

The bulk of all outliers (not shown) were caused by cloud cover and correctly removed. None of the outliers are seen to be correlated to detector position, as these should have been detected. Due to the orbit of TROPOMI, the observations are cyclical in nature ~every 16 days, with the same detector pixel observing the

same location. other local phenomena (e.g. sand storms, moisture content) were not checked.

sine wave correction: This is rather vaguely described to be honest. It would be good to show such a fit. Does it look better if fitted against AMF or SZA? Is this something that is also seen in MODIS data? This is interesting and curious but again, the authors chose to not go the extra mile, which would have made this paper much more interesting. In general I have no problem with not diving deeper into all the issues but given that the overall relevance of this manuscript for TROPOMI validation or validation schemes using PICS in general is rather thin, I would have expected a somewhat deeper analysis into these small curiosities.

The text was improved to better motivate the usage of the sine wave. Example fits are now given. However, its dependency on AMF/SZA, as well as occurrences in the MODIS data is under investigation. It is clear by now that it is not a simple detail, as discussed above. The sine wave appears to be caused by the BRDF effects and the yearly 'dependency' of solar angle.

What we thought was indeed a small curiosity is in fact a more complex problem, as it is not also clearly site-dependent. Using a site such as RailRoad Valley, where instrumentation is available to verify results likely gives a far more in-depth answer to the questions posed by the referee than performing this over remote sites such as in the Sahara.

**Line 131: "We attribute..." What is the basis of this, a hunch? You could actually look whether there is a VZA dependence! Why not do that, I really don't understand that choice.**

As detailed, this is due to the result in ongoing research. It is clear that a VZA dependence is present, but it definitely also is clear that it is not just a VZA dependence that can easily be corrected for. We have made a reference to future work in which we will go in-depth on these dependencies.

---

## Author Response (AR2)

This update addresses the proposed origin of the yearly variation to be the Earth-Sun distance.

This is not possible due to the time of perihelion near early January and the offset in days typically found for the sine wave fit. It also references the TROPOMI L1b ATBD.

---

## Author Response (AR3)

Reviewer comment:
The authors mostly addressed my comments, even though they appear hesitant to put more work into the current study.
* * *
Dear Referee
Thank you for the thorough and in-depth look. I hope I can address your points

With kind regards,

Tim
* * *
A few major points:

1) If there is a study in prep, please make sure the current manuscript won't become entirely obsolete. The authors should avoid just writing a separate paper that actually includes some of the things I would have liked to see in this one.
* * *
I agree with the author that I would like to have included a through and satisfying description on the BRDF effects as well. But this has proven, with the current knowledge and data availability, simply not realistic.
One of the main difficulties is the validation of any BRDF corrections. For this, one requires a site with ground instruments, such as RailRoad Valley, or other sites in the RADCALNET network (although for SWIR wavelengths currently only RRV suffices due to the lack of wavelength coverage).
Our investigations into this effect have progressed significantly, but cannot be applied to this paper. Primarily due to the absence of validation of these effects at any of the Saharn sites.
The work on RRV currently is extensive and requires its own paper.

There were three conclusions so far that are worth presenting in this response:
  - Over desert sites, the MODIS BRDF product was found not to be accurate enough. It often saturates over desert sites, in particular at longer wavelengths, providing a lack of data coverage. The uncertainties are also large.
  - VIIRS datasets are not saturated, but appear to lack the so-called 'hot spot' in their derivation, at least in the current publicly released dataset. Together with GOSAT and OCO we see consistent differences (see Bruegge et al. for the hotspot). A VIIRS dataset using the correct MODIS model would be a potential solution, but such a dataset is not publicly available.
  - Determining what/how the BRDF effects need to be corrected down a level below 5% in **absolute** terms has proven to be difficult. This is in line with the results as presented by previous work (Bruegge et al., 2019a/2019b, Kuze et al., 2014). 5% uncertainty is currently larger than the spread/variation we see. Note that we are aware we are comparing absolute (5% over RRV with atmospheric correction etc) and relative (3% spread) differences.

These results also directly show why the current paper will not be automatically be obsolete once the paper on RRV has been published. None of the sites given in this paper are instrumented. It thus cannot be validated that any BRDF effects derived for the RRV desert composition and applied to the Saharan, Namibian and Arabian deserts, is correct. This in particular is relevant at 2.3 micron where no other reliable instrumented site yet exists (at visible wavelengths one can compare RADCALNET sites).

Given the results of Bacour et al., the assumption that these effects are equal in RRV and any other desert is unlikely to hold down to the accuracies we have derived so far.
* * *
2) Can you be confident that BRDF is likely the main reason or can local heterogeneity (given the large footprint size) be a reason as well? As you haven't looked into MODIS yet, I doubt you can exclude the latter reason, right? Similarly for RR valley, Are there actually TROPOMI footprints that only covers the homogenous playa? PICS validation gets easier with finer spatial resolution, this should be clearly stated, esp. at the end when you make broad recommendations.
* * *
Local heterogeneity on scales of 20 and 100 km was investigated by Bacour et al., 2019 and is explicitly discussed there using the MODIS and Google Earth data. Heterogeneity has three possible effects, which can contribute:

- The effect would be seen by height heterogeneity (e.g. shadowing of dune formation), with the yearly variation. This has been mentioned. This is now expanded in the text as it is not fully explained.
- The pixel size of TROPOMI varies in size and would alter the BRDF by including different BRDF values at the edges. Any heterogeneity in the central part ($\sim$3 km$^2$) is not relevant as it is always included.
  If dominant, this would introduce a very regular pattern every $\sim$16 days due to the repeatability of the S5p orbit cycle (which consists of 227 orbits to be more precise). The pattern would be regular but complex in form, as the size difference of the TROPOMI pixel is not a linear function of swath position due to the curvature of the Earth, in addition of the location of the central reference location varying at a subpixel scale. The pattern's shape would be site-dependent, but its period would not.

  One can approximate whether or not this heterogeneity is influencing the results by looking at the Standard deviation we derive and compare that to the homogeneity values.
  By taking the Bacour et al., SHOM 20 km ($\sim$max TROPOMI pixel size) values of their Table 2 (using Optimal locations), one would expect the spread to be largest for Algeria3, Algeria5, Algeria2, Niger3, Arabia2, Namibia_PICSAND1 and Sudan_PICSAND1. These have larger SHOM values above 0.7%. Note that due to the swath position argument above, this number is very difficult to relate to TROPOMI results.

  However, Comparison with the STDEV of Table 3 of our paper shows there is no clear correlation for these sites with higher STDEV.
  This conclusion is reinforced by the results of the nadir view. If heterogeneity was a larger effect, the standard deviation would have to be severely reduced when only viewing the nadir view, even with the much lower number of available data points. This due to the pixel size being nearly identical for this subset. Although for some it is slightly reduced, this is also not seen.
  This analysis is included in the paper.
* * *
3) You mention 5 impact factors: (detector position, instrument zenith, instrument azimuth, solar azimuth, solar zenith. This actually points to the lack of atmospheric correction being a primary reason, or stray light or other effects? on-board calibration also has some drawback as the light doesn't fully the exact same path as on orbit during

regular observations. It would be good to discuss these pros/cons and clearly state the lack of atmospheric correction here as well.
* * *
The referee is correct that the lack of atmospheric correction can indeed be an effect, primarily due to the presence of aerosols and its variation. This is included in the papers assumptions and discussion.
Note that I hope that in my definition of atmospheric correction, the actual reflection on the surface is not included in this nomenclature. This is now included.

For various reasons, the other effects are all far less likely and have been discussed in earlier work.
TROPOMI-SWIR Straylight has been carefully studied both on-ground and in-flight (e.g. Tol et al., 2018, van Kempen et al., 2019, Ludewig et al., 2020) and has been shown to not be a factor for the SWIR detector. On-ground this was done over the identical optical path as the radiance (Tol et al., 2018). In-flight this is monitored by the diode lasers with variations seen less than tens of a percent (van Kempen et al., 2019, www.sron.nl/tropomi-swir-monitoring/swir-straylight). A similar argument can be made for the lack of variation in the ISRF inflight (van Hees et al., 2018).

Similarly, on-board calibration has shown the optics and detector to be stable down to tens of a percent or less (van Kempen et al., 2019). The only part of that is not actively monitored with either on-board lights or the irradiance, is the Earth-port entry/telescope itself (Figure 1 of van Kempen et al., 2019, green entrance at the lower right.). Long-term variation similar to what is seen over the Sahara deserts, is in itself monitored by reflectance monitoring over the entire Earth (monthly, see e.g. mps.tropomi.eu report, e.g.
http://mps.tropomi.eu/pdf/reporting?year=2021&number=07&freqchar=M&creationdate time=20210801T121718, page 12, see also Ludewig et al., 2020). No variations are seen down to levels of less than 0.5%. Spikes are produced by averaging more/less cloudy areas).

We hope this explains the points raised by the referee.

---

## Author Response (AR4)

Dear Editor,

Thank you for the time and effort of checking the paper.

I've have tried to address your comments as good as I could. In my opinion this works best by the addition of a new subsection (now 4.2) with a more thorough discussions on the limitations of the lack of BRDF corrections. The conclusions are updated to reflect that subsection. In my opinion, this structure is the best to present the conclusions given in my previous reply.

With kind regards

Tim van Kempen